# Resting-state brain metabolic fingerprinting clusters (biomarkers) and predictive models for major depression in multiple myeloma patients

**Xiaofei Wang** [1]*, **Joshua Eichhorn** [1], **Iqbal Haq** [1], **Ahmad Baghal** [2]

1 Department of Radiology, Nuclear Medicine Molecular Imaging and Target Therapy, University of Arkansas for Medical Sciences, Little Rock, AR, United States of America, 2 Department of Biomedical Informatics, University of Arkansas for Medical Sciences, Little Rock, AR, United States of America

* xwang@uams.edu

## Abstract

**Data Availability Statement:** All relevant data are within the paper and its Supporting Information files.

### Background

Major depression is a common comorbidity in cancer patients. Oncology clinics lack practical, objective tools for simultaneous evaluation of cancer and major depression. Fludeoxyglucose F-18 positron emission tomography–computed tomography (FDG PET/CT) is universally applied in modern medicine.

### Methods

We used a retrospective analysis of whole-body FDG PET/CT images to identify brain regional metabolic patterns of major depression in multiple myeloma patients. The study included 134 multiple myeloma (MM) patients, 38 with major depression (group 1) and 96 without major depression (group 2).

### Results

In the current study, Statistic Parameter Mapping (SPM) demonstrated that the major depression patient group (n = 38) had significant regional metabolic differences (clusters of continuous voxels) as compared to the non-major depression group (n = 96) with the criteria of height threshold T = 4.38 and extent threshold > 100 voxels. The five significant hypo- and three hyper-metabolic clusters from the computed T contrast maps were localized on the glass-brain view, consistent with published brain metabolic changes in major depression patients. Subsequently, using these clusters as features for classification learner, the fine tree and medium tree algorithms from 25 classification algorithms best fitted our data (accuracy 0.85%; AUC 0.88; sensitivity 79%; and specificity 88%).

**Funding:** The authors received no specific funding for this work.

**Competing interests:** The authors have declared that no competing interests exist.

## Conclusion

This study demonstrated that whole-body FDG PET/CT scans could provide added value for screening for major depression in cancer patients in addition to staging and evaluating response to chemoradiation therapies.

## Introduction

Major depression is one of the most common mental illnesses in the world. The Global Burden of Diseases, Injuries, and Risk Factors Study 2017 (GBD 2017) reported that depressive disorders moved up to one of the leading causes of YLD (years lived with disability) in 2017 with an estimated 264 million people suffering from depression [1]. Exacerbation of depression often happens in a setting of a comorbidity, particularly in cancer patients. Approximately 1 in 4 cancer patients experience major depression [2]. Patients with a history of depression are also more susceptible to relapse when they are diagnosed with cancer. Depressive disorders frequently compromise cancer treatments leading to increased mortality rates by up to 39% [3]. Several pathways may result in the symptoms that lead to the consideration of depression, including disruption in serotonin/dopamine pathways, the experience of loss or anticipated loss, direct side effects of chemotherapy medications, presence of tumors in the central nervous system, poorly managed pain, disruption of sleep due to medical treatments, and anemia [4].

Screening, assessing, and appropriately managing depression is essential in advanced cancer care. Simplified questionnaires, for instance, the Hospital Anxiety and Depression Scale (HADS), the nine-item Patient Health Questionnaire (PHQ-9), the Distress Thermometer, or the Impact Thermometer can be useful in screening for depression. However, the performance of these tools highly depends on professional experience and communication skills to make patients willing to share their feelings. An objective, reliable, and measurable method would be ideal for assessing a cancer patient's mental status. Positron emission tomography–computed tomography (PET/CT), a functional molecular modality, can assess cell and tissue functional changes [5], in particular with modern scanners, which have the capability of highly specific, accurately localized, reliable quantitative measurements of tissue concentration of tracers. Pioneer publications revealed brain regional metabolic changes between healthy subjects and patients with depression [6–8]. As new quantitative methodologies have been developed, for instance, Statistical Parametric Mapping (SPM) [9], there have been meta-analysis publications mapping metabolic changes of brain cluster voxels in major depressive disorder patients [10–13].

F-18 FDG PET/CT is increasingly playing a more critical role in staging, assessing treatment response, and surveillance, in particular multiple myeloma [14]. Tashiro et al explored metabolic changes of cancer patient's brains to reflect various psychological factors [15–17]. However, challenges remain evident since there are many confounding factors, including therapies [18, 19]. The University of Arkansas for Medical Sciences (UAMS) is one of leading Multiple Myeloma centers with the largest archived FDG PET/CT database. Multiple myeloma is a cancer, which derives from plasma cells, and accumulates in the bone marrow leading to bone pain and anemia. A survey by Lamers et al. showed that approximately 24% of multiple myeloma patients reported symptoms of depression [20]. To minimize the effects of confounder variables, we selected multiple myeloma patients with or without major depression from the FDG PET imaging database at the ratio of approximately one to three (depression vs. non-

depression). Using SPM and Matlab machine learning toolboxes, we identified clusters with a significant difference between the two groups and used the clusters as candidates of features for machine learning. In this current study, we explored predictive models to identify major depression cases from multiple myeloma patients.

## Materials and methods

### Subjects

The study's inclusion criteria included, using ICD 10 coding terminology, patients diagnosed with multiple myeloma. Clinical data, including demographics and comorbid medical conditions, were extracted from UAMS clinical data repository and confirmed by the patient's chart review. Using UAMS' FDG PET/CT imaging database from 2010 to 2019, we excluded patients with brain lesion or misregistration secondary to head motion during FDG PET image acquisition. The resulting study cohort included 138 patients that met the study criteria, divided into two groups. Group 1 (D) included 38 patients that have both multiple myeloma and major depression, and Group 2 (Control) included 96 patients that have multiple myeloma without major depression, (Table 1). This study was reviewed and approved by UAMS IRB (Protocol Number: 217785; PI: Xiaofei Wang).

### PET images

Following the procedure standard for tumor imaging with FDG PET/CT of the Society of Nuclear Medicine Molecular Imaging (SNMMI), patients fasted for at least 6 hours prior to the scan. The serum glucose level was checked before the FDG dose injection with the cutoff 200 mg/dL. Intravenous administration of 12–18 mCi (444–666Mbq) of $^{18}$F-FDG was performed in a quiet, warm, dimly lit room; however patients were not blindfolded. After approximately 60 minutes of uptake, a whole-body PET/CT scan was performed. Images from 2010 to 2015 were acquired on either a CTI-Reveal or a Biograph 6 PET/CT system (Siemens Medical Systems), both with full ring LSO crystal configurations (3 min/bed). PET images were generated by three-dimensional (3D) iterative reconstruction on a $168 \times 168$ matrix, with a zoom of 1.0, FWHM filter of either 5.0 or 6.0 mm, and two iterations with eight subsets. Images from 2015 to 2019 were acquired on Discovery IQ (GE Medical System) with 5-ring BGO-based detector blocks and 16-slice CT (2 min/bed). Images were reconstructed using two principle algorithms, VUE-point HD with point-spread-function modeling (VPHDS) and Q.Clear (QCHD) on a 192 x 192 matrix. CT data were used for anatomic localization and attenuation correction.

**Table 1. Demographic characteristics of multiple myeloma patients with and without depression.**

| Characteristic | Depression (n = 38) | Control (n = 96) | Statistical Significance (p value) |
|---|---|---|---|
| Sex (M:F) | 17:21 | 52:44 | 0.34[*] |
| Age (year) (mean ± SD) | 53 ± 7.4 | 55 ± 5.6 | 0.13[+] |
| Serum Glucose (mean ± SD) | 100.5 ± 21.9 | 101 ± 19.1 | 0.8[+] |
| Diabetes (Yes:No) | 4:34 | 9:87 | 1[*] |

Plus-minus values are mean ± standard deviation. The p-value was calculated with the use of a Fisher exact test[*] or unpaired t-test[+].

## Brain voxel based analysis

FDG brain images were extracted from the whole-body FDG PET/CT attenuation corrected images using Sectra stack tool, exported as DICOM files, and converted to analyze files (NifTI format) for subsequent post processing. The images were subjected to affine and nonlinear spatial normalization into the standard template of Montreal Neurological Institute (MNI) using SPM12 (Wellcome Department of Cognitive Neurology, London, UK). All default choices of statistical parametric mapping (SPM) were followed with dimension 91x109x91 (2x3 box: -90–126–72; 90 90 108), voxel size 2x2x2 mm, FWHM smooth 8x8x8 mm Gaussian filter to blur for individual variations in gyral anatomy and to increase the signal-to-noise ratio (Fig 1). The intensity normalization was performed by proportional scaling before obtaining T contrast maps. Each voxel intensity was first scaled by dividing each voxel value by the average of all the voxel values in the subject brain parenchyma using SPM12 to remove the confounding effects of changes in the global level with a masking threshold of 0.8. The adjusted voxel values were globally normalized to 50 ml/100 ml/min using proportional scaling. The resulting statistical parametric maps, SPMT, were generated using flexible factorial design. A threshold of $p < 0.05$ (voxel-level) family-wise error (FWE)–corrected for multiple comparisons and a minimum cluster size of 100 was applied to all analyses. Using Marsbar [21], templates of

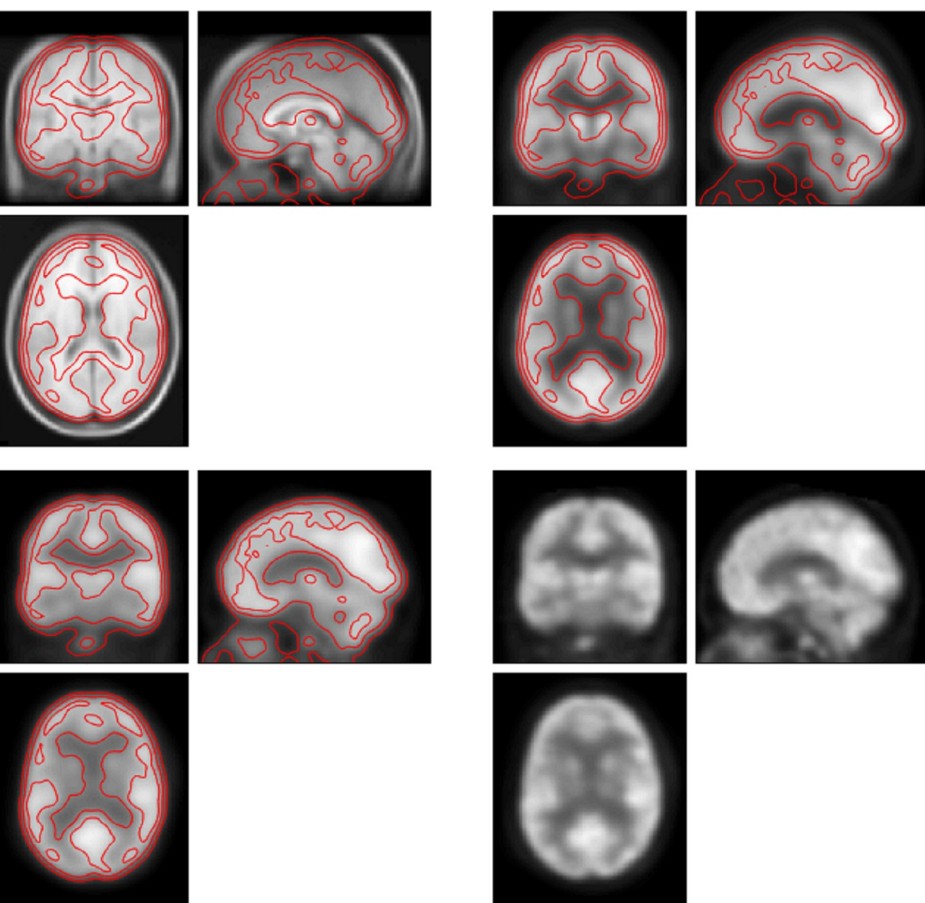

**Fig 1. Example of FDG PET imaging post-processing.** Left upper panel: MRI brain template (T1) with coronal, sagittal, and axial views, generated from 305 normal subjects. Right upper panel: FDG PET brain template of summed 100 normal subjects. Left lower panel:Smoothed FDG PET brain of one study subject. Right lower panel: Normalized FDG PET brain of the same subject. The red line is the contour of the subject's normalized brain imaging.

cluster ROIs were created if clusters met the threshold. Signals were extracted from each image within ROIs as clusters for further machine learning.

### Machine learning

We applied a supervised machine learning approach with 5-fold cross-validation to determine which classification algorithms would fit our data with the best performance [22]. The clusters with statistical significance would serve as candidates of features for model comparisons. Classification Learner automatically employed twenty-five algorithms to fit our data with different feature selection combination. Each model's performance was measured by accuracy, confusion matrix, and receiver operating characteristic (ROC)-area under curve (AUC).

## Results

### Global metabolic difference between multiple myeloma with or without major depression

We measured the global average FDG activity of each subject, and further compared the difference of means between two groups. From Fig 2, group 1 (Depression) has significantly lower global FDG activity in comparison to group 2 (Control) (p < 0.001 and Cohen's d -1.33). The box plot also shows marked variation among individuals of brain FDG uptake as expected from our experience.

### Clusters with significant metabolic changes

Voxel-based analysis reveals significant regional metabolic changes in major depression patients. Globally, there are extensive brain regions (5831 voxels) demonstrating hypo-

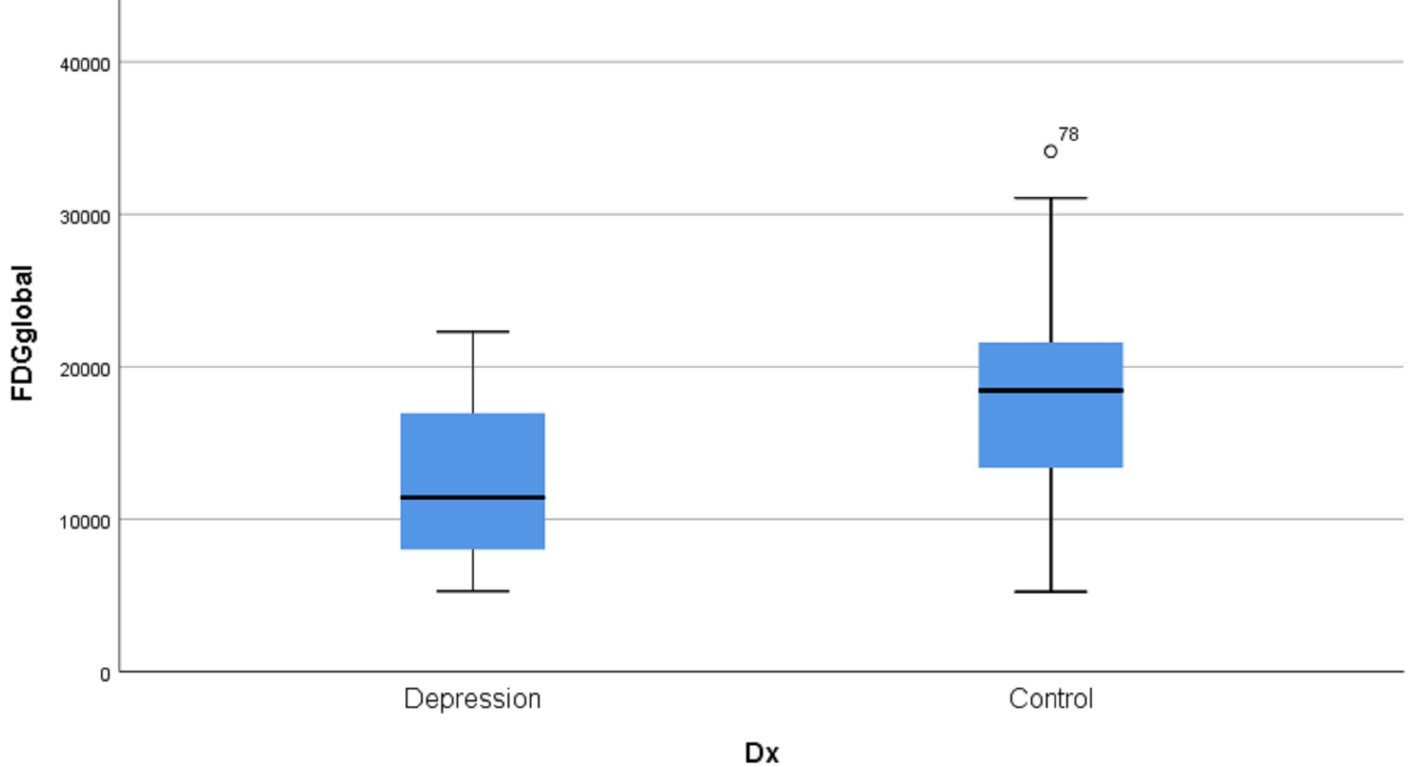

**Fig 2. Box plot of global FDG activity in major depression group (D) and control group (N).**

**Table 2. Significantly changed metabolic clusters corresponding anatomic/functional regions.**

| AAL Region | TD Labels | BA (large to small) | Cluster (voxels) | T (Peak) | Z (Peak) | Peak Coordinates (x, y, z) |
|---|---|---|---|---|---|---|
| Frontal_Mid_R | Middle Frontal Gyrus | 9, 46, 8, 10, 6, 44, 45 | 1740 | -6.69 | -6.2 | 44, 32, 28 |
| Frontal_Mid_L | Middle Frontal Gyrus | 9, 46, 6, 10, 8, 44, 45 | 1459 | -6.2 | -5.79 | -46, 24, 26 |
| Angular_R | Angular Gyrus | 40, 39, 19, 2, 3, 18, 7, 1 | 1247 | -6.08 | -5.7 | 44, -64, 36 |
| Parietal_Inf_L | Supramarginal Gyrus | 40, 39, 19, 7 | 760 | -5.89 | -5.54 | -52, -50, 38 |
| Temporal_Mid_R | Middle Temp Gyrus | 21, 20, 37, 22, 19 | 625 | -5.39 | -5.11 | 62, -40, -8 |
| ParaHippocampal_L | Uncus | 28, 34, 36 | 696 | 5.78 | 5.45 | -24, 0, -32 |
| Brainstem_R | | | 183 | 4.92 | 4.71 | 14, -26, -22 |
| Amygdala_R | Sub-Gyral | | 215 | 4.82 | 4.62 | 30, 2, -16 |

AAL: automated anatomical labeling. TD Labels: automated Talairach atlas labeling. BA: Brodmann area (areas with at least five voxles are listed). Signs of T or Z value: negative indicates hypo-metabolic changes; positive denotes hyper-metabolic changes.

metabolism (Group 1 < Group 2) in patients with major depression (Table 2). To a lesser extent, regions of hyper-metabolism (Group 1 > Group 2) are appreciated as well (1094 voxels) (Table 2). To better visualize the metabolic changes, the results of the SPM T maps were displayed on xjView [23]. Subsequently, we confirmed the corresponding anatomic locations of each peak MNI of these significant hyper- or hypo-metabolic clusters in major depression patients. Fig 3 illustrates these clusters on Collin-27 T1 MRI brain template. Interestingly, hyper-metabolic clusters were anatomically localized in limbic system bilaterally and right brainstem (Fig 3 and Table 2), whereas hypo-metabolic clusters predominantly involved bilateral frontal, parietal, and right temporal lobes (Fig 3 and Table 2).

## Machine learning

We used eight clusters as features without Principal Component Analysis (PCA) and hyper-parameters during machine learning and explored twenty-five classification algorithms. Results are shown on Table 3, Figs 4 and 5. Overall Fine Tree and Medium Tree performed the best (accuracy 85.1%; ROC-AUC 0.88; sensitivity 79%; and specificity 88%).

## Discussion

Major depression is a severe mental disorder and its mechanism remains not well understood in spite of decades of rigorous research. Based on our limited knowledge, major depressive disorder is complicated, temporary or permanent neuron function and/or interaction changes between neurons and glial cells, and/or physical structure changes based on our limited knowledge [24–26]. Molecular functional imaging makes in vivo brain research feasible and translates finding for facilitating diagnosis of disorders and assessment of therapy. Brain perfusion SPECT/CT imaging showed regional brain perfusion defects in depression patients, particularly lateral frontal lobe and temporal lobe [27–31]. FDG PET/CT metabolic imaging, another surrogate of brain functional activity, correlates to blood flow and local energy metabolism [32]. As a quantitative molecular imaging modality, the exploratory studies in late 19th and early 21st century revealed metabolic changes in multiple areas of major depression patients as compared to healthy control subjects. However, there were discrepancy of anatomic locations and hyper/hypo-metabolic changes among published FDG PET/CT studies due to heterogeneity of clinical syndrome and genetics of depression [33, 34], and relative small sample sizes and ROI definitions published [11, 13].

To our knowledge, this is the first study to explore whole brain metabolic changes between major depression and non-major depression among multiple myeloma patients. The study

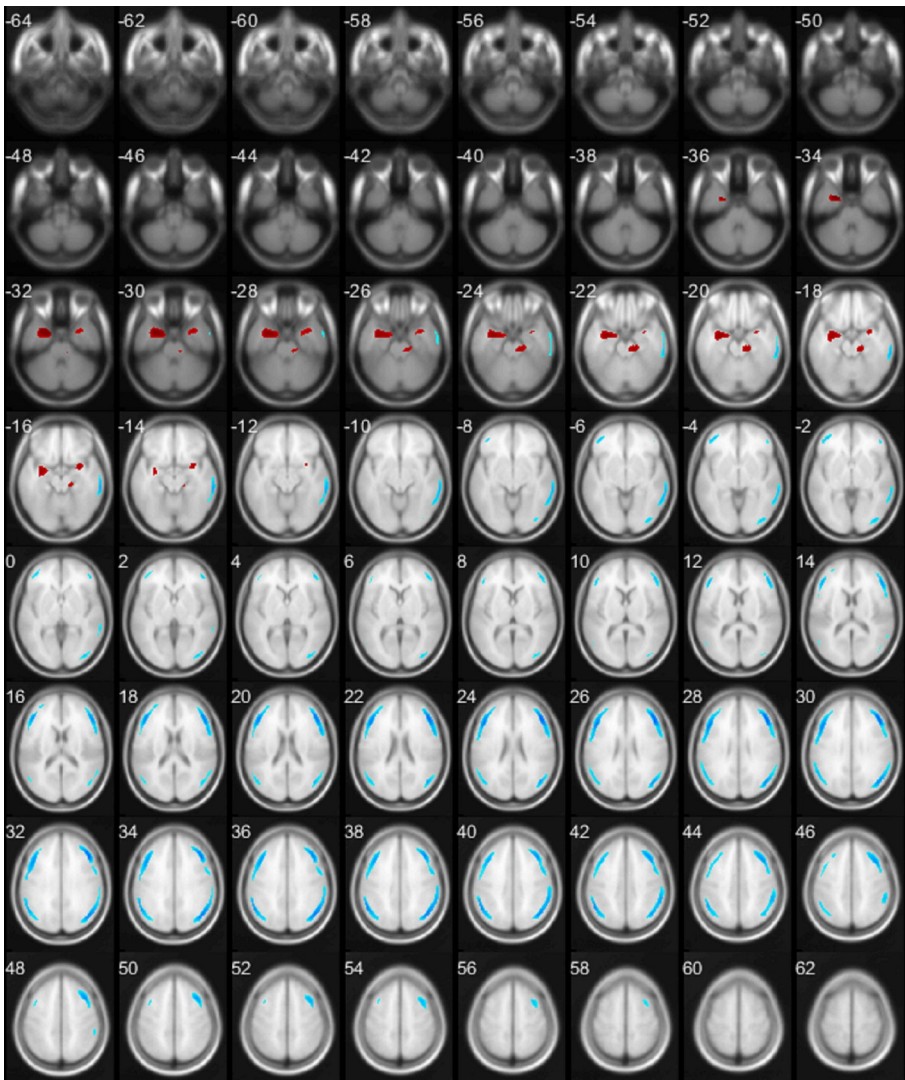

**Fig 3. A map of significantly changed metabolic clusters in patients with major depression as compared to the control.** Statistical parametric maps displayed on transverse sections depicting regions of hyper-metabolic (red) and hypo-metabolic (blue) changes.

demonstrated that the major depression group had a significant reduction of metabolic activity as compared to the non-major depression group at the global level (Cohen's d -1.33). Secondly, using voxel-based analysis, we identified 5 hypo- and 3 hyper-metabolic clusters above the pre-set threshold of $p < 0.05$ (FWE) and greater than 100 voxels in multiple myeloma patients with major depression. Thirdly, we validated two predictive models with for screening major depression among multiple myeloma patients using these fingerprinting clusters as classification features.

## Global hypo-metabolism in multiple myeloma with major depression

Our data revealed a significant global hypo-metabolic change in major depression. Global hypo-metabolic changes in depression have been previously observed [35, 36]. From our data, multiple myeloma patients with major depression presented a -1.33 of Cohen's d, absolute

**Table 3. Performance of 25 classification algorithms.**

| Model ID | Model Name | 8ROIs_accuracy | 8ROIs_AUC | Sensitivity | Specificity |
|---|---|---|---|---|---|
| 1 | **Fine Tree** | **85.1** | **0.88 (0.13, 0.79)** | **79** | **88** |
| 2 | **Medium Tree** | **85.1** | **0.88 (0.13, 0.79)** | **79** | **88** |
| 3 | Coarse Tree | 80.6 | 0.86 (0.19, 0.79) | 79 | 81 |
| 4 | Linear Discriminant | 78.4 | 0.81 (0.09, 0.47) | 47 | 91 |
| 5 | Quadratic Discriminant | 76.9 | 0.72 (0.13, 0.5) | 50 | 88 |
| 6 | Logistic Regression | 76.9 | 0.8 (0.09, 0.42) | 42 | 91 |
| 7 | Gaussian Naïve Bayes | 79.1 | 0.85 (0.18, 0.71) | 71 | 82 |
| 8 | Kernel Naïve Bayes | 79.1 | 0.85 (0.18, 0.71) | 71 | 82 |
| 9 | Linear SVM | 82.1 | 0.82 (0.09, 0.61) | 61 | 91 |
| 10 | Quadratic SVM | 79.1 | 0.73 (0.09, 0.50) | 50 | 91 |
| 11 | Cubic SVM | 73.9 | 0.74 (0.17, 0.50) | 50 | 83 |
| 12 | Fine Gaussian SVM | 71.6 | 0.73 (0.00, 0.00) | 0 | 100 |
| 13 | Medium Gaussian SVM | 80.6 | 0.80 (0.09, 0.55) | 55 | 91 |
| 14 | Coarse Gaussian SVM | 80.6 | 0.85 (0.08, 0.53) | 53 | 92 |
| 15 | Fine KNN | 76.9 | 0.70 (0.14, 0.53) | 53 | 86 |
| 16 | Medium KNN | 79.1 | 0.81 (0.15, 0.63) | 63 | 85 |
| 17 | Coarse KNN | 71.6 | 0.78 (0.00, 0.00) | 0 | 100 |
| 18 | Cosine KNN | 76.9 | 0.82 (0.22, 0.74) | 74 | 78 |
| 19 | Cubic KNN | 81.3 | 0.81 (0.13, 0.66) | 66 | 88 |
| 20 | Weighted KNN | 77.6 | 0.82 (0.13, 0.53) | 53 | 88 |
| 21 | Boosted Trees | 71.6 | n/a | 0 | 100 |
| 22 | Bagged Trees | 75.4 | 0.78 (0.16, 0.53) | 53 | 84 |
| 23 | Subspace Discriminant | 80.6 | 0.82 (0.08, 0.53) | 53 | 92 |
| 24 | Subspace KNN | 76.9 | 0.80 (0.15, 0.55) | 55 | 85 |
| 25 | RUSBoosted Trees | 77.6 | 0.83 (0.20, 0.71) | 71 | 80 |

ROIs: regions of interest. AUC: area under curve.

value larger than patients with HAM-D scores 22 or above (-0.83) in Kimbrell's report. Our data indicated that multiple myeloma might worsen major depression.

The decrease in FDG uptake in major depression indicates less glutamate neurotransmission through glucose oxidation process, which links to major depression [37–40]. Our data suggested that multiple myeloma patients with major depression might have extensive interruption of glutamatergic transmission. It has been estimated that as much as 70% of the energy derived from glucose oxidation is required to convert glutamate, the major neurotransmitter in the brain, to glutamine during glutamatergic transmission. Therefore, the FDG PET signal is proxy for measuring excitatory neurotransmission [37, 41].

## Fingerprinting clusters (biomarkers) in multiple myeloma with major depression

Whole brain voxel-based analysis identified five significant hypo-metabolic and three significant hyper-metabolic clusters in multiple myeloma patients with major depression. Our data showed that the summed volume of hypo-metabolic clusters is much larger than that of the hyper-metabolic clusters, which logically explains the observed global hypo-metabolic activity seen.

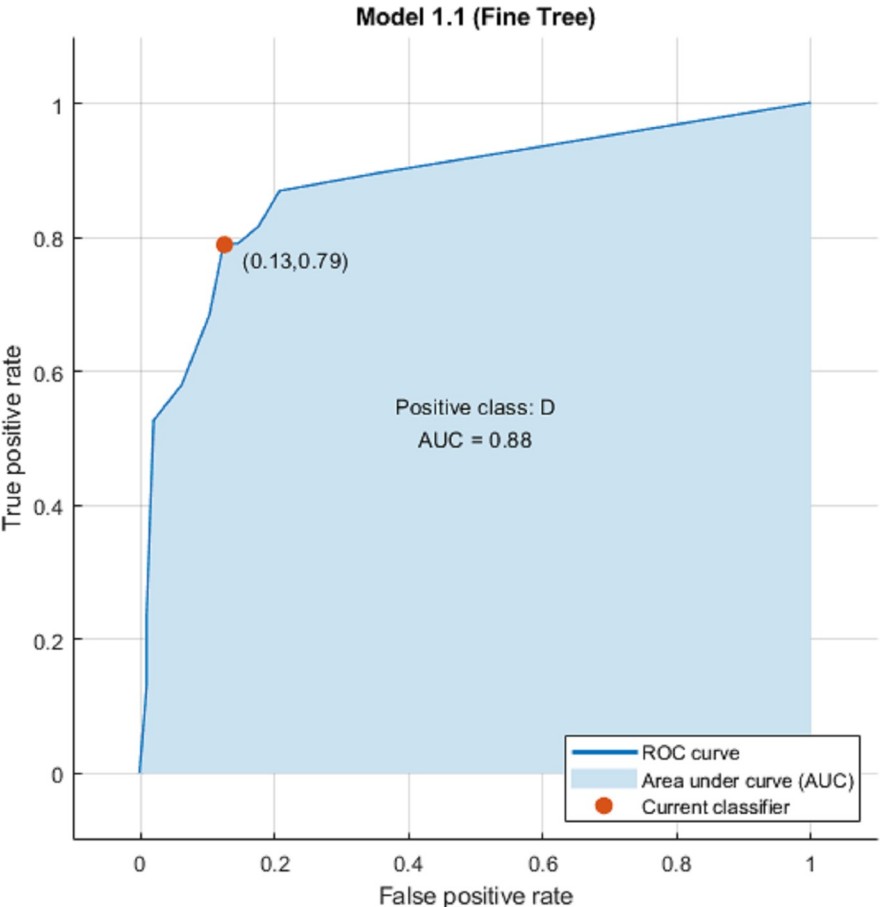

**Fig 4. Fine tree model ROC-AUC.**

The two largest hypo-metabolic clusters with peak MNI coordinate 44, 32, 28 (x, y, z) and -46, 24, 26 (x, y, z) predominantly locate in bilateral middle frontal gyrus, contiguously extending to the adjacent frontal cortex. The hypo-metabolic changes in the frontal lobes were noted in major depression patients in previous studies [42–48].

The third and fourth hypo-metabolic clusters with peak MNI coordinate 44, -64, 36 (x, y, z) and -52, -50, 38 (x, y, z) correspond to bilateral angular gyrus and supramarginal cortex, also involving part of the neighboring occipital and temporal cortices. The reduction of metabolic activity in these brain regions have previously been reported in major depression patients [44, 48–50].

The fifth hypo-metabolic cluster with peak MNI coordinate 62, -40, -8 (x, y, z) sits in the right middle and inferior temporal gyri. The right temporal area with less FDG uptake has also previously been observed in patients with major depression [42, 43, 48].

The largest hyper-metabolic cluster with peak MNI coordinate -24, 0, -32 (x, y, z) grossly maps to the left limbic lobe, mainly involving in the left parahippocampal gyrus, and adjacent areas of amygdala and hippocampus. Patients with major depression without anxiety have been shown to have significantly increased FDG uptake in left parahippocampal gyrus as compared to normal subjects [50]. Of note, the left amygdala had less FDG uptake in the patients with a suicide plan [44]. Interestingly, hypometabolic changes in depressed unipolar patients have previously been reported in the anterior cingulate of limbic lobe, differing from our

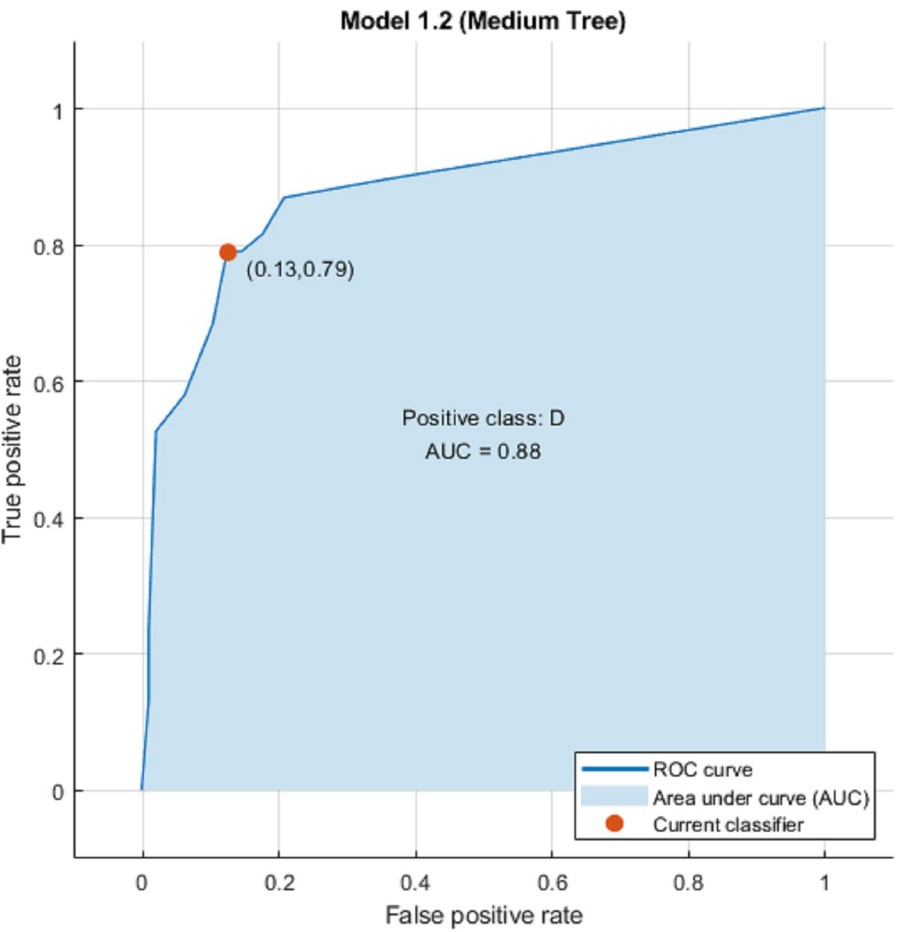

**Fig 5. Medium tree model ROC-AUC.**

findings and others'[13, 35, 47, 51]. However, unlike this and other studies, subjects in Kimbrell et al study performed an auditory continuous performance task (CPT) 30 min during the FDG uptake period.

The second most significant hyper-metabolic cluster with peak MNI 14, -26, -22 (x, y, z) is seen in the right sided brainstem and partially involves the adjacent right cerebellum, anterior lobe and parahippocampal area. The third hyper-metabolic cluster with peak MNI 30, 2, -16 (x, y, z) is visualized in the right amygdala. The last two clusters are close anatomically but separated under our current threshold. Higher FDG activity in brainstem (global maximum at Talairach coordinates of 2, −24, −20) was seen in non-remitters of major depression patients after three-month monoaminergic medication as compared to remitters, suggestive of potential prognosis for the enhancing anti-depression treatment [52]. In cancer patients, distress thermometer (DT) and DT problem list positively correlated to FDG activity in brain stem [53].

We compared our SPM T-maps to previous studies' T-maps or equivalent maps available. Based on the literature provided cluster selection criteria, we found that the pattern of metabolic changes showed on Fig 5 was consistent with previous studies [36, 43, 48, 52, 54–56]. Interestingly, hypo-metabolic clusters always presented larger T scores and volume/size than hyper-metabolic ones. Therefore, if cluster height threshold was set too large, hyper-metabolic clusters in major depression would not been seen. Milak et al studies reported metabolic

correlation with psychopathologic factors derived from the self-rated Beck Depression Inventory (BDI), factors from the clinician-rated Hamilton Depression Rating Scale (HDRS) factors, and verbal learning deficit in major depressive disorder [52, 57, 58]. There is high degree of anatomic overlap among clusters, which is consistent with metabolic change directions noted in our results and Milak et al. reports, suggestive of the metabolic changes correlating to major depression symptoms in multiple myeloma patients. As noted, the metabolic clusters in our data are the essential components reported in putative networks related to depression, further indicating aberrant connectivity of ventral limbic affective, frontal-striatal reward network, default mode network, and dorsal cognitive control network [59].

### Predictive models for screening major depression among multiple myeloma

This study is first to use machine learning algorithms to identify presence of major depression in multiple myeloma patients using FDG PET/CT. Among the 25 machine learning methods tested, the fine tree and medium tree models from our data showed the best performance with 85.1% of accuracy, 79% of sensitivity, and 88% specificity. Comparing to previous resting-state fMRI and perfusion SPECT machine learning studies with large samples (n > 100) [60–63], our results are comparable to the better classification performance among the studies (89.2% of overall accuracy).

**Limitations.**   The present study is a retrospective analysis of patients with diagnoses of multiple myeloma and major depression. There was no significant difference in demographic characteristics between the Control and Depression groups regarding age, gender, blood glucose levels, and diabetes. However, potential confounding variables, such as current treatment regimens, coexisting anxiety disorders or other combidities, and depressive symptoms at the time of the scan were not taken into account and represent possible biases. Furthermore, no feasible control group could be utilized as healthy patients do not usually undergo PET/CT. Additionally, analysis was performed on standard whole-body FDG PET/CT exams without dedicated imaging of the brain. While this may pose limitations to image quality, the results demonstrated here are more widely applicable to patients undergoing standard FDG PET/CT for oncologic evaluation.

### Conclusion

In this study, we demonstrate that oncology FDG PET/CT images can provide useful information for screening for comorbid major depression in cancer patients without extra dedicated brain images or increased radiation exposure or cost. This would not replace the need for a psychological evaluation however this may alert the clinician to metabolic changes in the brain that have been associated with depression. The patient may have sub-clinical depression or may have been embarrassed to express their symptoms to their oncologist. This would serve as a supplemental screening exam to the more traditional questionnaires. The eventual progression of these methods is to screen all patients undergoing PE/CT for major depression. With the rapid advancement of AI, there is potential for automated screening of patients during the staging PET/CT for major depression, and this may increase awareness and identification of patients needing additional support for depression.

### Supporting information

**S1 Fig. A Map of significantly changed metabolic clusters (T = 3.15) in patients with major depression as compared to the control.** Statistical parametric maps displayed on transverse

sections depicting regions of hyper-metabolic (red) and hypo-metabolic (blue) changes. (TIF)

## Acknowledgments

We thank Dr. Hanna Jensen for editorial assistance.

## Author Contributions

**Conceptualization:** Xiaofei Wang.

**Data curation:** Xiaofei Wang.

**Formal analysis:** Xiaofei Wang.

**Investigation:** Xiaofei Wang.

**Methodology:** Xiaofei Wang, Joshua Eichhorn, Iqbal Haq.

**Supervision:** Xiaofei Wang.

**Validation:** Xiaofei Wang.

**Visualization:** Xiaofei Wang.

**Writing – original draft:** Xiaofei Wang.

**Writing – review & editing:** Joshua Eichhorn, Ahmad Baghal.

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
