## [Decision Letter · Decision Letter 0]

10 Mar 2021

PONE-D-20-36796

Resting-state brain metabolic fingerprinting clusters (biomarkers) and predictive models for major depression in multiple myeloma patients

PLOS ONE

Dear Dr. Wang,

Thank you for submitting your manuscript to PLOS ONE. After careful consideration, we feel that it has merit but does not fully meet PLOS ONE’s publication criteria as it currently stands. Therefore, we invite you to submit a revised version of the manuscript that addresses the points raised during the review process.

We look forward to receiving your revised manuscript.

Kind regards,

Matteo Bauckneht

Academic Editor

PLOS ONE

Journal Requirements:

Reviewers' comments:

Reviewer's Responses to Questions

**Comments to the Author**

1. Is the manuscript technically sound, and do the data support the conclusions?

Reviewer #1: Yes

Reviewer #2: Yes

Reviewer #3: Partly

2. Has the statistical analysis been performed appropriately and rigorously? 

Reviewer #1: Yes

Reviewer #2: Yes

Reviewer #3: N/A

3. Have the authors made all data underlying the findings in their manuscript fully available?

Reviewer #1: Yes

Reviewer #2: Yes

Reviewer #3: No

4. Is the manuscript presented in an intelligible fashion and written in standard English?

Reviewer #1: No

Reviewer #2: Yes

Reviewer #3: Yes

5. Review Comments to the Author

Reviewer #1: The manuscript faces an interesting, though not commonly observed, topic. Here below there are some suggestions and some corrections that should be made, in my opinion, in order to make it eligible for publication.

Errors:

- Line 57: please substitute "(Satin 2009)" with the progressive reference number and then modify the list of refs.;

- Line 62: as above, please replace the link with the reference number;

- Line 74: please correct "have developed" with "have BEEN developed";

- Line 78: please repeat the acronym "PET/CT" in place of the full name;

- Line 86: did you mean "approximately 24% of multiple myeloma PATIENTS suffered from"?

- Line 161: please replace the link with the reference number;

- Line 235: please put all the reference numbers together.

Aspects that should be better clarified:

- M&M section: the modality of acquisition of brain PET images must be explained. For example, was it a separate bed acquisition? What was the duration of the brain scan? Did you use any immobilization system to avoid motion artifacts? How did you orient patient's head?

- Results Section: please describe more in detail the areas that have shown to differ from group 1 and 2;

- Line 224: please explain the link between glutamate neurotransmission, FDG uptake and major depression;

- Please add a paragraph/a few sentences about the limits of your study.

Reviewer #2: Thank you very much for an interesting paper regarding the use of FDG PET for predicting major depression in cancer patients. I appreciate the retrospective use of PET data for assessing additional clinical use of the functional metabolic data.

I will not address the postprocessing of the scans, since this is not my field of expertise. I will rather comment on the differences between the two groups.

On a general note to the changes observed in brain glucose metabolism in the depression group: could the findings also/additionally be explained by possible patient anxiety or maybe more likely by ongoing medication/chemotherapy?

And for that matter, are data for patient diabetes, BS levels/fasting compliance available? The insulin levels are very important for global and regional uptake of FDG. Also where the patients blindfolded/in a quiet room during the uptake period?

Are there known differences in morbidity between the groups except the major depression?

F.ex other mental illnesses that could mimic depressive metabolic reductions in the cortex of the brain.

I would prefer a few comments considering these above mentioned circumstances, preferably addressing them as possible biases.

Regarding the conclusions. I would prefer a few lines addressing the possible use of the conclusions achieved. Could you imagine all brain PET data from cancer patients going through a screening for major depression using your predictive model?

Would it not be more feasible to recommend a psychological evaluation of all patients and the PET data as an add on?

And a side note: In our PET department the brain is not scanned in cancer patients on a regular basis to spare radiation to the lens of the eyes. In your experience, how many FDG PET scans in cancer patients include the brain as well?

More specific comments:

P. 5, l. 100: Were the 96 patients screened for depression?

P. 12, l. 220: I would like this statement explained. Did the cancer patients with major depression worsen? And did patients without major depression develop it afterwards?

P.12, l. 222: This is a repeated statement from p.11.

P.13, l. 243 In ref 11, the limbic lobe was a hypo-metabolic area in major depression patients. In your data it is a hyper metabolic cluster, what could be the reason for this?

Reviewer #3: The paper in general terms is understandable but it is recommended to have some additional edition because there are some points in the methodology and results that were not completely clear ; the methodology, as mentioned, is not completely clear and the “discussion” the seems very general to me and not well elaborated (Authors focus on the explanation of their results rather than providing a bunch of possible explanations. Why do the authors choose this machine learning algorithms ? ).

Materials and Methods

Subjects:

The authors mentioned ‘FDG PET/CT images from 138 patients met the criteria. Thirty-eight patients with

100 major depression as group 1 (D), and 96 patients without major depression as group 2 (Control)’. It is not very clear what are diagnostic criteria for major depression group and without major depression group?

It is unreasonable that you draw a conclusion only with the two groups. A healthy control group should be added.

PET images:

The data are from different PET/CT system ,and reconstructed using different method.It is known that PET data are affected by these differences, esecially for voxel-wised analysis. it would be taken in consideration in the SPM analysis.

Discussion:

Though the potential interest in Major depression has been previously reported, the authors do not clearly state what the potential relevance of Major depression among multiple myeloma patients. This should be stated more clearly so that the motivation of the study is readily apparent to the reader.

6. PLOS authors have the option to publish the peer review history of their article (what does this mean?). If published, this will include your full peer review and any attached files.

Reviewer #1: **Yes: **Priscilla Guglielmo

Reviewer #2: No

Reviewer #3: No

---

## [Author Response · Author response to Decision Letter 0]

12 Apr 2021

I have submitted the document entitled with Response to Reviewers, point-by-point response to the questions and comments delivered in your letter on 3/10/2021

---

## [Editor Report · Decision Letter 1]

19 Apr 2021

Resting-state brain metabolic fingerprinting clusters (biomarkers) and predictive models for major depression in multiple myeloma patients

PONE-D-20-36796R1

Dear Dr. Wang,

We’re pleased to inform you that your manuscript has been judged scientifically suitable for publication and will be formally accepted for publication once it meets all outstanding technical requirements.

Kind regards,

Matteo Bauckneht

Academic Editor

PLOS ONE
---

## [Editor Report · Acceptance letter]

26 Apr 2021

PONE-D-20-36796R1 

Resting-state brain metabolic fingerprinting clusters (biomarkers) and predictive models for major depression in multiple myeloma patients 

Dear Dr. Wang:

I'm pleased to inform you that your manuscript has been deemed suitable for publication in PLOS ONE. Congratulations! Your manuscript is now with our production department. 

Kind regards, 

on behalf of

Dr. Matteo Bauckneht 

Academic Editor

PLOS ONE